# Preoperative Mixed-Reality Visualization of Complex Tibial Plateau Fractures and Its Benefit Compared to CT and 3D Printing

**DOI:** 10.3390/jcm12051785

**Published:** 2023-02-23

**Authors:** David Bitschi, Julian Fürmetz, Fabian Gilbert, Maximilian Jörgens, Julius Watrinet, Robert Pätzold, Clemens Lang, Claas Neidlein, Wolfgang Böcker, Markus Bormann

**Affiliations:** 1Department of Orthopedics and Trauma Surgery, Musculoskeletal University Center Munich (MUM), University Hospital, LMU Munich, 81377 Munich, Germany; 2Department of Trauma Surgery, Trauma Center Murnau, 82418 Murnau, Germany

**Keywords:** AR/MR glasses, MR visualization, 3D printing, tibial plateau fracture (TPF)

## Abstract

Background: Various studies have shown the benefit of three-dimensional (3D) computed tomography (CT) reconstruction and especially 3D printing in the treatment of tibial plateau fractures (TPFs). This study aimed to investigate whether mixed-reality visualization (MRV) using mixed-reality glasses can provide a benefit for CT and/or 3D printing in planning treatment strategies for complex TPFs. Methods: Three complex TPFs were selected for the study and processed for 3D imaging. Subsequently, the fractures were presented to specialists in trauma surgery using CT (including 3D CT reconstruction), MRV (hardware: Microsoft HoloLens 2; software: mediCAD MIXED REALITY) and 3D prints. A standardized questionnaire on fracture morphology and treatment strategy was completed after each imaging session. Results: 23 surgeons from 7 hospitals were interviewed. A total of 69.6% (*n* = 16) of those had treated at least 50 TPFs. A change in fracture classification according to Schatzker was recorded in 7.1% of the cases and in 78.6% an adjustment of the ten-segment classification was observed after MRV. In addition, the intended patient positioning changed in 16.1% of the cases, the surgical approach in 33.9% and osteosynthesis in 39.3%. A total of 82.1% of the participants rated MRV as beneficial compared to CT regarding fracture morphology and treatment planning. An additional benefit of 3D printing was reported in 57.1% of the cases (five-point Likert scale). Conclusions: Preoperative MRV of complex TPFs leads to improved fracture understanding, better treatment strategies and a higher detection rate of fractures in posterior segments, and it thus has the potential to improve patient care and outcomes.

## 1. Introduction

Tibial plateau fractures (TPFs) are complex injuries of the knee joint with increasing incidence [1,2,3]. The greatest increase is seen in postmenopausal women, suggesting osteoporosis as a key factor for TPFs [3]. That is why TPFs in elderly women are already classified as “major osteoporotic fractures” [4,5]. Due to the aging population, a further increase in incidence is expected. Based on X-ray images in two planes, Schatzker et al. published a classification system for TPFs in 1979, which is still used today. They described six types of TPFs based on patient age, bone quality, fracture morphology and trauma energy. Schatzker I-III fractures involve only the lateral tibial plateau and are caused by low-energy trauma. High-energy trauma is the cause of Schatzker IV-VI fractures. A Schatzker IV fracture is a fracture of the medial tibial plateau only. In contrast, fractures of the medial and lateral tibial plateau with or without discontinuity of the metaphysis are found in Schatzker V and VI fractures [6]. Since the description of TPFs by Schatzker et al. in 1979, several studies have shown that the complexity of TPFs is often not captured in two-dimensional imaging and in particular, fracture components in the posterior segments are missed. These studies were based on computed tomography (CT) scans of the injured knee joint. With these CT images, the fractured tibial plateau could be viewed in multiple planes and sections for the first time [7,8,9,10,11,12]. With its increasing availability, the CT scan became the gold standard in radiological diagnostics of TPFs [7,13,14]. The availability of magnetic resonance imaging (MRI) has also increased significantly, but in contrast, the standardized use of the preoperative MRI examination is still controversially discussed in treatment of TPFs [15,16,17]. Based on CT, new classification systems developed, and 38 different ones are currently described in the literature. Based on the fracture classification, some of these classification systems recommend a specifical surgical treatment strategy [18]. The improvement in fracture detection using CT contributed to the development of new surgical approaches and new implants, which led to the 360° treatment of the tibial plateau used today. This 360° treatment requires a detailed understanding of the fracture morphology and includes open and arthroscopic osteosynthesis procedures [7,10,11,12,19,20,21,22,23]. The intra- and inter-observer reliability of the classification systems have also improved with the standardized use of preoperative CT imaging [24].

Development from two-dimensional (2D) to three-dimensional (3D) fracture analysis continues. For TPFs, 3D reconstruction of the CT data is already recommended [14,19,25]. Three-dimensional printing technology has further improved preoperative three-dimensional fracture visualization. In addition to a three-dimensional view of the fracture, the printed fracture model also provides a haptic perception of the fracture. For the treatment of fractures in general and specifically for TPFs, preoperative 3D printing shows a benefit in fracture understanding and surgical treatment. These benefits are evident in a reduced operation time, a reduced complication rate and a more accurate reconstruction [26,27,28,29,30,31,32,33]. However, there is no standardized use of this technology to date.

The latest development in 3D CT imaging is mixed-reality visualization (MRV) on mixed-reality (MR) glasses. There are important differences between virtual reality (VR), mixed-reality (MR) and augmented reality (AR) glasses. While VR glasses shield the user completely from the outside world, MR and AR glasses allow interaction with the physical world by superimposing a digital image or generated 3D model (hologram) over the user’s view. The main difference between MR and AR glasses is that MR offers the possibility to directly interact with the holograms, whereas AR limits the experience to visualization. MR glasses are constantly evolving, so there have been improvements in comfort, expansion of the field of view, resolution and even precision and handling [34,35,36]. Compared to 3D printing, an advantage of the MRV technique is that is offers an immediate visualization without any printing time. In orthopedics/trauma surgery, studies demonstrated a benefit of MRV compared to CT. The benefits can be seen in the preoperative planning as well as during the operation [36,37,38]. However, these studies do not evaluate MRV compared to 3D printing. Furthermore, they commonly refer to its use in elective joint replacement. The few studies on the use of MRV in trauma refer to the intraoperative use of MR glasses [36]. In these studies, MR glasses are used for the navigated insertion of single screws [39,40,41,42], for example, in sacral fractures or as locking screws. All these studies so far have been exclusively experimental in vitro studies. Studies on the use of MRV in TPFs are lacking.

This study evaluates the preoperative MRV of complex TPFs concerning fracture understanding and treatment strategy, as well as its value compared to CT and 3D printing.

## 2. Materials and Methods

CT data of three complex TPFs from a German Level I trauma center were selected for this study and prepared for 3D visualization. Subsequently, the fractures were presented to participating specialists in trauma surgery by use of CT, MR glasses and as 3D prints, each with a standardized questionnaire on fracture morphology and treatment strategy.

Intra-articular, bicondylar TPFs with anterior and posterior fracture components were required. The CT device (Discovery CT750 HD—GE Healthcare, Chicago, USA) and CT protocol (slice thickness <0.7 mm) used were identical for all three fractures. The CT data set was prepared for MRV and 3D printing using software from the company mediCAD (mediCAD 6.8 with the module 3D Knee Sport 2.1.19—mediCAD Hectec GmbH, Altdorf, Germany). For 3D printing, manual adjustments were made in the software. Printing was carried out using an FDM printer (Raise3D N2 Plus—Raise 3D Technologies, CA, USA) out of ABS plastic (Formfutura TitanX). MRV was performed on a Microsoft HoloLens 2 (hardware: Microsoft HoloLens 2—Microsoft Corporation, Redmond, WA, USA; software: mediCAD MIXED REALITY MR 1.1—mediCAD Hectec GmbH, Altdorf, Germany).

Each TPF was presented in three steps to the participating surgical specialists in trauma surgery. First by CT, including 3D CT reconstruction (software: Visage 7.1.16—Visage Imaging, San Diego, CA, USA; Figure 1), subsequently on MR glasses (Figure 2), and finally as a 3D-printed model (Figure 3).

After each presentation step, subjects were given the same standardized questionnaire on fracture morphology, treatment strategy and subjective perceived certainty regarding fracture understanding and selected treatment strategy (see the Appendix A). The questionnaire was created with the web application SoSci Survey (SoSci Survey GmbH, Munich, Germany).

Furthermore, the participating specialists were asked about the number of independently surgically treated TPFs. The processing of the first two fractures was obligatory, while participation in the processing of the third fracture was optional.

Statistical analysis was performed using SPSS Statistics 26.0 software (IBM Corp., Armonk, NY, USA). Wilcoxon and McNemar tests were chosen for statistical analysis, with a significance level of *p* < 0.05. Graphical representation was performed using Microsoft Excel 365 MSO version 2207 (Microsoft Corp., Redmond, WA, USA).

The study was approved by the local ethics committee (21-0559) and complies with the Declaration of Helsinki ethical standards.

## 3. Results

A total of 23 surgeons from 7 hospitals (3× care level 3, 2× care level 2, 1× care level 1, 1× specialist hospital) were included in this study. Altogether, 69.6% (*n* = 16) of the interviewed surgeons had surgically treated at least 50 tibial plateau fractures (10–50 TPFs: 30.4% (*n* = 7), 50–100 TPFs: 52,2% (*n* = 12), >100 TPFs: 17.4% (*n* = 4)). None of the participants had used MRV in clinical practice prior to the study.

Out of the 23 participants, 10 answered the questionnaire for all three fractures and the remainder for two fractures. This resulted in 56 cases/evaluations.

Preoperative MRV had no significant influence on additionally desired preoperative MRI imaging (desired in 42.9% (*n* = 24) after CT). After MRV visualization, the desire changed in 1.8% (*n* = 1) of cases compared to the CT. There were no changes after the presentation of the 3D print.

### 3.1. Fracture Classification

A change in Schatzker classification after MRV compared to CT viewing was recognized in 7.1% (*n* = 4) of the cases. There was no further change after presentation of the 3D print. All three fractures were most frequently classified as Schatzker type 5: 55.4% (*n* = 31) after CT and 60.7% (*n* = 34) after MRV and 3D prints (Figure 4).

Based on the ten-segment classification, MRV led to a change in the selected segments in 78.6% (*n* = 44) of cases. After presentation of the 3D print, there were further changes in 42.9% (*n* = 24). Figure 5 shows the frequency of the selected segments after each presentation step.

A tendency towards more frequently selected posterior segments after MRV and 3D printing was observed.

### 3.2. Treatment Strategy

Primary treatment changed in 3.6% (*n* = 2) of cases following MRV (external fixator instead of a Mecron splint). Table 1 provides a further overview of changes in the planned definitive treatment of TPFs.

An increased number of changes in intraoperative patient positioning following MRV was observed (51.8% (*n* = 29) with CT only and 62.5% (*n* = 35) following MRV).

Planned treatment by a posterior approach increased following MRV (83.9% vs. 92.9%). Combined approaches as well as isolated treatment via a posterior approach increased in prevalence after MRV (10.7% vs. 19.6%).

Planned osteosynthesis was more frequently combined with a posterior plate following MRV than with CT only (87.5% vs. 94.6%). Furthermore, additional screw osteosynthesis was planned more often (64.3% vs. 73.2%).

### 3.3. Subjective Results

Based on five-point Likert scale (agreement or complete agreement), a benefit of the MR glasses compared to CT was seen in 82.1% (*n* = 46) of the cases. Another benefit of 3D printing was seen in 57.1% (*n* = 32) of the cases.

Perceived certainty in the understanding of fracture morphology increased significantly after MRV compared to CT (*p* < 0.001). Another non-significant increase was seen after reviewing the 3D print (*p* = 0.132; Figure 6).

## 4. Discussion

This study shows a subjective and objective benefit of preoperative MRV for the treatment of complex TPFs. Compared with CT, this benefit is similar to that of a 3D-printed fracture model.

More than 30 different classification systems for TPFs have been described in the literature [18]. This raises the question: What is the purpose of a classification system? According to Audigé et al., a fracture classification system should contribute to a better understanding of fracture morphology, improved communication between clinicians, easier documentation and better treatment decision making [43]. However, precise and reproducible fracture classification is not only important in pre- and intraoperative settings. It is also of great importance for the comparability of treatment strategies and outcome in TPFs. Therefore, a classification system contributes significantly to the improvement of fracture patients. It has already been shown that the intra- and interrater reliability of established classification systems for TPFs are higher with CT imaging than with conventional X-ray imaging [13]. A recent study demonstrated that the intra- and interrater reliability of the AO/OTA, revisited Schatzker and ten-segment classification systems was significantly increased by the use of 3D printing compared to CT with 3D reconstruction [26]. Our study shows changes in the Schatzker classification with higher agreement after MRV and no further changes after considering 3D printing.

The detection and treatment of posterior fracture segments, which are frequently missed in two-dimensional representations, are essential for today’s established 360° treatment and influence patient outcome [7,19,44,45,46,47]. The worse outcome here is partly due to mechanical instability if the posterior segments are not adequately fixed [7,48,49,50]. In addition, posterior segment injuries appear to be associated with a higher incidence of ligament and meniscus injuries [51]. If overlooked or untreated, these injuries could contribute to a poorer outcome for the patient. Therefore, in our own procedure, MRI imaging is performed in TPFs with involvement of the posterior segments. This study demonstrates that with MRV, fractures in posterior segments are detected and treated more frequently.

Furthermore, 3D CT reconstruction allows 3D viewing of the fractured tibial head, but on a 2D computer screen. Several studies have demonstrated the superiority of 3D-printed fracture models compared to 3D CT reconstruction for treatment of TPFs [26,27,28,29,30,31,32,33]. The benefits of 3D printing are demonstrated by shorter operating times, less intraoperative blood loss, less intraoperative fluoroscopy, lower complication rates, faster fracture healing and better Rasmussen and HSS outcome scores [31]. Despite these advantages, preoperative 3D printing has not yet been established as a standard for the treatment of complex TPFs. Possible reasons for this are the cost- and time-intensive production of 3D-printed models [31,52]. The CT data must first be converted by software for the preparation of the print. Due to the often multi-fragmentary damage at the tibial plateau, at least part of the sequencing must be conducted manually. The time this requires adds up along with the printing time of several hours, and manual finishing of the model is often necessary. In addition to these indirect costs, the direct costs result from the purchase and maintenance of the 3D printer as well as the costs of the printing material.

MRV of the CT data on the MR glasses is carried out by the software within seconds, without any required manual editing. The initial investment for MR glasses and its software is, in our case, roughly equivalent to the costs of purchasing and maintaining a 3D printer for one year.

In this study, MRV provides results comparable to 3D printing in terms of perceived certainty in understanding the fracture morphology and the chosen treatment strategy. Differences in the selected treatment strategy after MRV and after 3D printing are evident, although the clinical relevance of these cannot be clarified in this work.

In our study, a benefit of MRV was reported in 82.1% of the cases, and a further benefit of 3D models was reported in 57.1% of cases. The missing haptic in MRV seems to have a significant influence on this. Furthermore, it must be considered that all participants had no previous experience with the use of MR glasses. Like any new technology, there is a learning curve in the use of MR glasses. The learning curve mainly relates to faster handling of the MR glasses. It is conceivable that routine use of MR glasses will result in the use of all visualization features of MR glasses, which may further improve the understanding of fracture morphology. It remains to be clarified whether the effect of missing haptics decreases along the learning curve.

Individual studies in arthroplasty and spine surgery already demonstrate the possibility of virtual planning and intraoperative navigation using MR glasses [36,37,38]. For TPFs, studies show the benefit of individualized reconstruction planning on the 3D-printed fracture model [29,53,54]. To enable surgeons to develop a patient-specific surgical strategy, a further development of the existing software and hardware is essential, including virtual reduction and osteosynthesis on the MR glasses. In this context, an algorithm for structured automated fragment integration and alignment has already been described in the literature [55]. Furthermore, a continued development of MRV navigated osteosynthesis, which has so far been limited to single screws in in vitro experiments, would be preferable. These developments would allow surgeons to design a precise preoperative plan and implement it intraoperatively.

We focused on complex TPFs, as the literature had previously shown an advantage of 3D printing mainly for these fractures [26,30,32]. However, in simple fractures, posterior fragments may also be missed, and the depth of the impression may be misjudged. Especially in simple fractures, these factors influence the indication for conservative or surgical treatment. The increased incidence of TPFs in the elderly population is mainly due to low-energy trauma with an increase in Schatzker II and III fractures [3]. An individualized treatment strategy is particularly important in this group of patients due to concomitant comorbidities. A benefit of MRV, therefore, also seems conceivable for these fractures.

It must be clarified by further studies to what extent the mentioned advantages of MRV in the preoperative management of complex TPFs contribute to a better outcome for the patients.

## 5. Conclusions

Preoperative MRV of complex TPFs leads to increased certainty in fracture understanding and planned treatment strategy and a higher detection rate of fractures in posterior segments and thus has the potential to improve patient care and outcomes.

## 6. Limitations

This study was limited by the small number of fractures and participants and the lack of randomization of fracture presentation. Another limitation was the presentation of the fracture three times, which inevitably led to a more intensive study of the fracture.

## Figures and Tables

**Figure 1 jcm-12-01785-f001:**
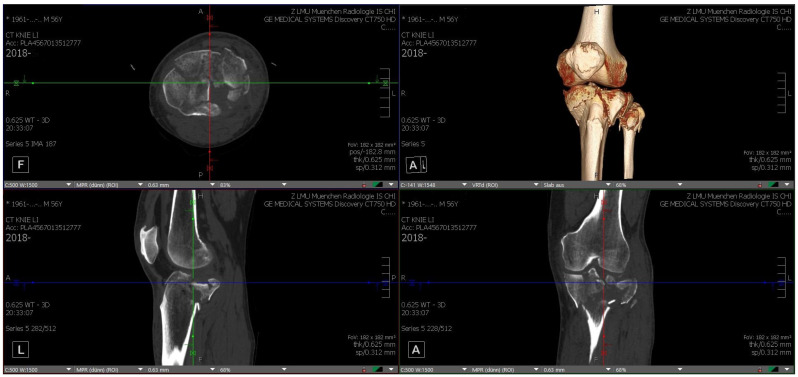
CT (axial, sagittal, coronary) including 3D reconstruction of fracture 1 (source: Visage 7.1.16—Visage Imaging, self-modified).

**Figure 2 jcm-12-01785-f002:**
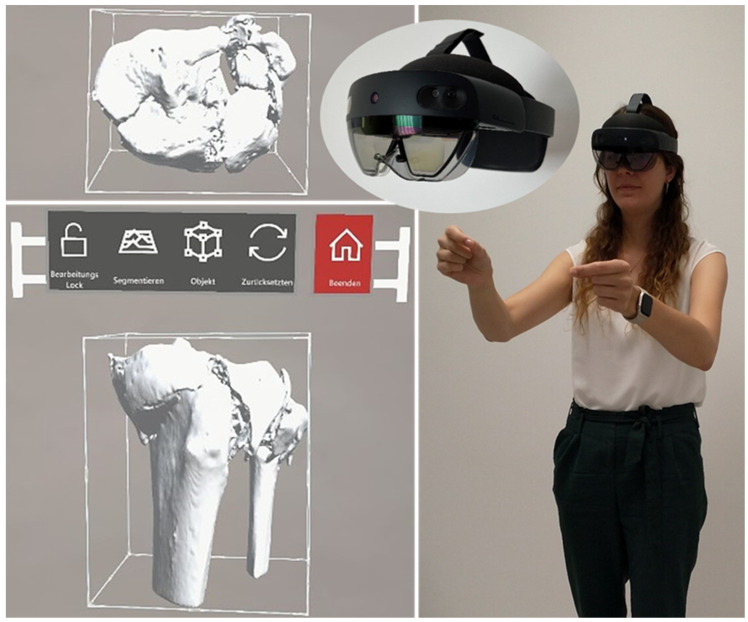
Mixed-reality glasses with holograms (axial and frontal) of fracture 1.

**Figure 3 jcm-12-01785-f003:**
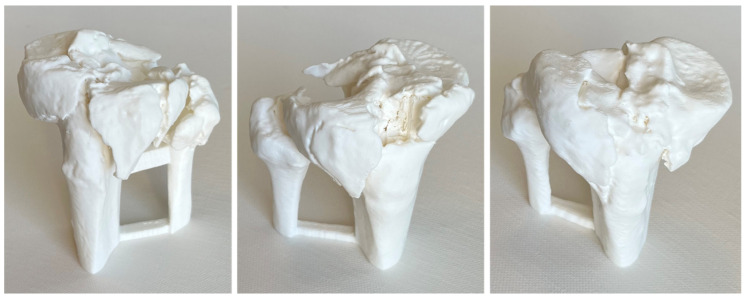
Three-dimensional-printed model of fractures 1–3.

**Figure 4 jcm-12-01785-f004:**
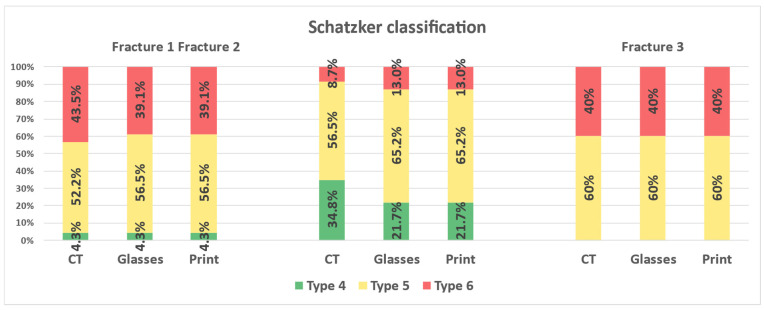
Schatzker classification of the three fractures.

**Figure 5 jcm-12-01785-f005:**
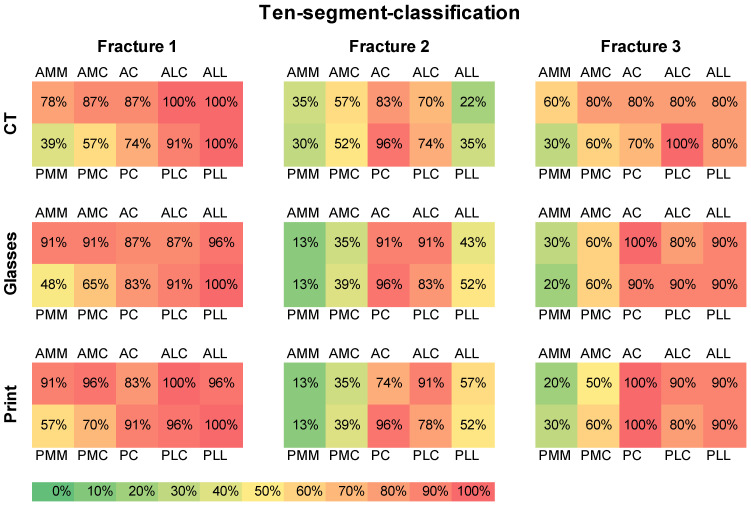
Ten-segment classification of the three fractures with frequency distribution of the segments.

**Figure 6 jcm-12-01785-f006:**
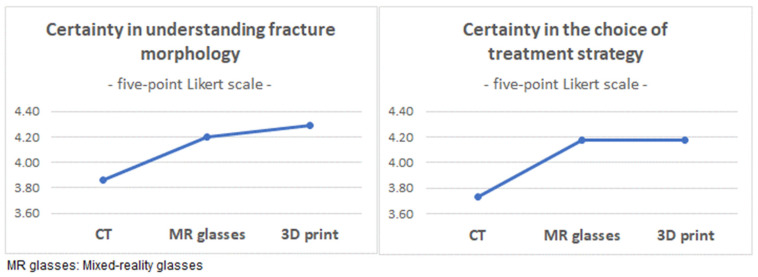
Certainty in understanding of the fracture morphology and choice of treatment strategy.

**Table 1 jcm-12-01785-t001:** Overview of changes in the planned definitive treatment.

Change in	Following MRV in % of Cases (*n* = Number)	Following 3D Print in % of Cases (*n* = Number)
Positioning	16.1% (*n* = 9)	5.4% (*n* = 3)
Approach	33.9% (*n* = 19)	19.6% (*n* = 11)
Approach extension	8.9% (*n* = 5)	3.6% (*n* = 2)
Osteosynthesis	39.3% (*n* = 22)	25% (*n* = 14)
Intraoperative reduction control	10.7% (*n* = 6)	5.4% (*n* = 3)

MRV: Mixed-reality visualization.

## Data Availability

Not applicable.

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
