# Peer review of "Preoperative Mixed-Reality Visualization of Complex Tibial Plateau Fractures and Its Benefit Compared to CT and 3D Printing"

_jcm, 2023, doi:10.3390/jcm12051785_

Round 1
Reviewer 1 Report
The manuscript is very interesting, introducing all surgeons to the new world of visual reality models.
The choice of fractures of Schatzker type 4-6 is appropriate, but it is reasonable that the use of 3D printing and MRV will change the classification even in types 1-3, where possible change of treatment strategy is even more important.
Can you please comment in discussion for the change in initial classification even for the simple tibial plateau fractures ( Schatzker 1-3)
Learning curve of the surgeons involved was minimally evaluated, as they had no previous experience. In more experienced ones, did the authors expect even HIGHER rate of changes in evaluation of fractures?
Reviewer 2 Report
The research aims to investigate whether mixed reality visualization using mixed reality glasses-12 can provide a benefit to computed tomography and/or three-dimensional printing in planning treatment strategies for complex Schatzker Type 4, 5, 6 tibial plateau fractures.
The introduction synthetically approaches the problem of the study and finally formulates the objective of the research.
In the Materials and Methods paragraph, the stages of the study methodology, which are continuously covered, could be presented synthetically in a first sentence.
The results of the questionnaire distributed to the specialists could be represented graphically for a better understanding of their opinion.
In the discussion section, the implications of this study should be interpreted regarding the perspectives of fracture fragment alignment by referring to some results presented in the literature, e.g. Moldovan, F.; Gligor, A.; Bataga, T. Structured Integration and Alignment Algorithm: A Tool for Personalized Surgical Treatment of Tibial Plateau Fractures. J. Pers. Med. 2021, 11, 190. https://doi.org/ 10.3390/jpm11030190.
I suggest that the titles of the bibliographic references in German, from ex poz 12,18, should be translated into English between brackets.
There some editing errors as follows:
In lines 73, 74, 75 - Error! Reference source not found – should be replaced with Fig.1,2,3.
In lines 110-112, there is a text from template that should be deleted because it is not related to the work.
Line 117 - (Error! Reference source not found.) should be replaced with Fig 4.
Line 123 - Error! Reference source not found should be replaced with Fig 5.
Line 152 - Error! Reference source not found. should be replaced with Fig 6.
Overall, the work is innovative and I congratulate the team of authors for their work.
